# Magneto Rotational Instability in Magnetized AGN Tori

**Yuki Kudoh \* and Keiichi Wada** 

Graduate School of Science and Engineering, Kagoshima University, Kagoshima 890-0065, Japan; wada@astrophysics.jp

\* Correspondence: k5751778@kadai.jp

**Abstract:** It is widely believed that, in active galactic nuclei (AGNs), a supermassive black hole with an accretion disk is surrounded by an optically and geometrically thick torus at sub-parsec scale. However, it is not clear how the mass supply is toward the central engine caused and how it is related with the internal structures of the tori. The magnetic field in the tori may contribute to the accretion process via the magneto-rotational instability (MRI). Using global three-dimensional magnetohydrodynamic (MHD) simulations taking the effects of X-ray heating and radiative cooling into account studied the numerical resolution for azimuthal direction for MRI driving. We found that a strongly magnetized disk consisted of a cold ($<10^3$ K) and warm ($10^4$ K) gas is developed in about 30 rotational periods. We also found in a high resolution model that the mean azimuthal magnetic fields reverse their direction quasi-periodically. We confirmed that the typical wave length of the MRI should be resolved with a least 20 azimuthal grid cells.

**Keywords:** galaxies: nuclei; methods: numerical; magnetohydrodynamics (MHD)

## 1. Introduction

Accretion processes onto central objects are important for evolution of the astrophysical objects; e.g., the X-ray binary, Gamma ray burst, and active galactic nuclei (AGNs). Since the most accreted gases have angular momentum, angular momentum transport is a problem. Turbulent viscosity was suggested by Shukura and Sunyaev [1], but the driving source of the turbulence was not identified. Balbus and Hawley [2] pointed out that the magneto-rotational instability (MRI) can account for the angular momentum transport in the differentially rotational magnetized disks. Mass accretion and angular momentum transport in magnetized gas disks can be described as follows. Small amplitude perturbation of the magnetic field lines in the radial direction are stretched and amplified for the radial direction by MRI growth. Since magnetic field lines are frozen into the ionized gas under the MHD approximation, deforming magnetic fields transport the gas and the angular momentum to central region. A viscosity proportional to the gas pressure was conventionally assumed in the equation of motion, but it is replaced by Maxwell's stress, $B_r B_\varphi$.

MRI-driven turbulence amplifies the magnetic field strength in the linear regime, and drives a buoyancy force due to Parker instability [3]. Nonlinear evolution of MRI with the Parker instability is characterized by a quasi-periodic reversal of the direction of azimuthal field in spacetime diagrams (e.g., [4–7]). The typical growing timescale of MRI is the rotational timescale determined by a balance of radial direction between the gravity, centrifugal force and magnetic tension, and the period of the quasi-periodic reversal is about 10–20 rotational periods. Therefore, a long-term calculation beyond 10 rotational periods are required in order to study the time evolution of nonlinear evolution of MRI and the mass transfer due to the MHD turbulence.

Here, we focus on sub-pc structures of the magnetized gas around a supermassive black hole (SMBH). The accretion rate in the AGNs is related to their luminosities as

$$\dot{M} = \frac{L}{\eta c^2} \sim 0.2 \left[ \frac{M}{10^7 \, M_\odot} \right] M_\odot \text{yr}^{-1}, \tag{1}$$

where $L$ is the bolometric luminosity, $c$ is the light speed, and $\eta$ is efficiency of the energy conversion, respectively. The SMBHs are surrounded by optically and geometrically thick tori (see, e.g., [8,9]), and the accreted material toward SMBH could be originated in the tori. Magnetic fields penetrating the torus were reported by the mid-infrared spectro-polarimetry of NGC1068 and NGC4151 [10,11]. Therefore, for sake of understanding the torus accretion, MRI is one of the mechanisms of the angular momentum transport. Dorodnitsyn and Kallman [12] and Chan and Krolik [13] carried out the simulations of magnetized AGN tori. However, the nonlinear MRI has been insufficiently understood. It is not clear whether MRI drives with cold gas in AGN tori.

We have studied the MRI in the cold gas of AGN torus using global three-dimensional MHD simulations including the X-ray heating and the radiative cooling. We investigate how the azimuthal resolution depends on driving MRI in the cold dense gas using the high-order numerical scheme.

## 2. Methods

We used CANS+ code [14], which is implemented with the HLLD method [15] and the hyperbolic divergence cleaning method [16]. In addition, the 5th order spatial accuracy is achieved by the monotonicity preserving method (MP5 [17]). The high order interpolation requires reducing the numerical diffusion of magnetic fields for a long-term calculation. A cylindrical coordinate $(r, \varphi, z)$ is used in the computational domain, $0 < r \, [\text{pc}] \, < 11, 0 < \varphi < 2\pi$, and $|z| \, [\text{pc}] > 3$. The numbers of grid points are $(N_r, N_z) = (256, 512)$, and we chose $N_\varphi = 128$ or $512$. For the outer region ($|z| > 2.8$ [pc], $r > 10$ [pc]) and the central region ($\sqrt{r^2 + z^2} < 0.4$), the outflow boundary conditions and absorbing boundary conditions are used, respectively.

We solve the following MHD equations:

$$\frac{\partial \rho}{\partial t} + \boldsymbol{\nabla} \cdot [\rho v] = 0, \tag{2}$$

$$\frac{\partial}{\partial t} (\rho v) + \boldsymbol{\nabla} \cdot \left[ \rho vv + \left( P_\text{g} + \frac{B^2}{8\pi} \right) \boldsymbol{I} - \frac{\boldsymbol{BB}}{4\pi} \right] = -\rho \boldsymbol{\nabla} \Phi, \tag{3}$$

$$\frac{\partial}{\partial t} \left( \frac{P_\text{g}}{\gamma_\text{g} - 1} + \frac{1}{2}\rho v^2 + \frac{B^2}{8\pi} \right) + \boldsymbol{\nabla} \cdot \left[ \left( \frac{\gamma_\text{g}}{\gamma_\text{g} - 1} P_\text{g} + \frac{1}{2}\rho v^2 \right) v - \boldsymbol{E} \times \boldsymbol{B} \right] \\ = -\rho v \cdot \boldsymbol{\nabla} \Phi + \rho L, \tag{4}$$

$$\frac{\partial \boldsymbol{B}}{\partial t} = \boldsymbol{\nabla} \times \boldsymbol{E}, \tag{5}$$

$$\boldsymbol{E} = v \times \boldsymbol{B} - \eta \boldsymbol{\nabla} \times \boldsymbol{B}, \tag{6}$$

where $\rho$, $P_\text{g}$, $v$, $\boldsymbol{B}$ is the gas density, pressure, velocity vector, magnetic field, respectively. In order to evaluate the temperature, we assumed the ideal gas with $\gamma_\text{g} = 5/3$. Electric field, $\boldsymbol{E}$, is related as the Ohm law, and the magnetic resistivity $\eta$ adopts the the anomalous resistivity model (e.g., [5,18]). This is in effect where the magnetic reconnection occur and regulated to be $\eta < 10^4 (r/1 \, [\text{pc}])(v/207 \, [\text{km/s}])$

[cm$^2$/s]. The gravitational potential $\Phi$ is $GM/\sqrt{r^2 + z^2}$, where $M$ is the mass of SMBH, $10^7 M_\odot$. We ignore the self-gravity of the gas.

We combined the radiative cooling and heating effects into Equation (4),

$$\rho L = n \left( \Gamma_{\text{UV}} + \Gamma_{\text{Coulomb}} \right) + n^2 \left( \Gamma_{\text{Compton}} + \Gamma_{\text{photoionic}} - \Lambda \right), \tag{7}$$

where $n \, (= \rho / m_{\text{H}})$ is the number density, and $m_{\text{H}}$ is the mass of neutral hydrogen. Cooling function was modeled on Wada et al. [19] and Wada [20] with the solar abundances. We take X-ray and UV from the accretion disk into account as heating processes. $\Gamma_{\text{UV}} = 1.8 \times 10^{-25} [\text{erg s}^{-1}]$ is assumed. X-ray heating [21] due to Compton interaction and photoionization for $10^4 < T < 10^8$ are:

$$\Gamma_{\text{Compton}} = 8.9 \times 10^{-36} \xi \left( T_{\text{X}} - 4T \right) [\text{erg s}^{-1}\text{cm}^3], \tag{8}$$

$$\Gamma_{\text{photoionic}} = 1.5 \times 10^{-21} \xi^{1/4} T^{1/2} \left( 1 - T/T_{\text{X}} \right) [\text{erg s}^{-1}\text{cm}^3], \tag{9}$$

where $T_{\text{X}} = 10^8$ K is the characteristic temperature of X-ray. The Coulomb heating is

$$\Gamma_{\text{Coulomb}} = \eta_{\text{h}} H_{\text{X}} [\text{erg s}^{-1}]. \tag{10}$$

$\eta_{\text{h}}$ denotes the efficiency [22,23], which assume to be fixed 0.2, and $H_{\text{X}}$ is X-ray energy deposition rate $H_{\text{X}} = 3.8 \times 10^{-25} \xi$ erg s$^{-1}$. The X-ray luminosity in the nucleus is parameterized by ionization parameter,

$$\xi = \frac{L_{\text{X}}}{n \left( r^2 + z^2 \right)} \exp(-\tau) \, [\text{erg cm s}^{-1}], \tag{11}$$

where $L_{\text{X}}$ is X-ray luminosity and $\tau$ is the optical depth, respectively. We assumed $\xi = 1.0$ for simplification.

The initial condition is expressed as the superposition of dynamical equilibrium solution with the isothermal spherical symmetry and the weakly magnetized hot torus (see, [5,24]). We adopted that the torus center is at 1 [pc] and rotation velocity is $v_\varphi = 207(r/1 \, [\text{pc}])^{-0.65}$ [km/s] for the torus and $v_\varphi = 0$ otherwise. The initial magnetic field in the torus is $(B_r, B_\varphi, B_z) = (0, P_g/\beta, 0)$, and plasma beta is 100 defined as $\beta = 2P_g / \left( B_r^2 + B_\varphi^2 + B_z^2 \right)$. The simulations used the normalized unit, i.e., $l_0 = 1$ [pc] for the length, $v_0 = \sqrt{GM/l_0} = 207$ [km/s] for the velocity, $n_0 = 10^2$ [cm$^{-3}$] for the number density, and $B_0 = \sqrt{4\pi m_{\text{H}} n_0 v_0^2} = 3.3$ [mG] for the magnetic field, respectively.

We initially evolve the model adiabatically, and once the MRI-driven turbulence is fully developed and becomes quasi-stable at about 20 rotational periods at $r = 1$ [pc], the cooling and heating terms are taken into account.

## 3. Results

Figure 1 from left to right shows time evolution in the high resolution model ($N_\varphi = 512$): (1) initial state, (2) the adiabatic phase where the MHD turbulence is developed, and (3) the cooling/heating phase, respectively. Initial torus of (1) constitutes the high temperature ($10^4 < T < 10^{5.3}$) gases; (2) shows that the gases are heated by Joule heating and spread out the radial and vertical direction from initial torus. The radial spread of gas is a result of the angular momentum transport with the deformation of magnetic field lines attracted by MRI. The vertical spread is caused by the magnetic field to the vertical direction. After the heating and cooling tern on (Figure 1(3)), a geometrically thin disk consisted of cold gas ($<10^3$ K), which is surrounded by a warm gas halo ($\sim 10^{4-5}$ K) is formed.

The magnetic field of the $rz$ plane slice is shown in Figure 2: (a) $B_\varphi$ and (b) plasma beta. As shown in Figure 2(a2), a turbulent structure is developed due to MRI. It is also notable that the turbulence

consists of opposite directions of $B_\varphi$ represented by blue and red regions. The plasma beta decreases to order unity from the initial value ($\beta = 100$). After cooling and heating are taken account, the spatial sign reversal of $B_\varphi$ is dissipated and formed a stripe-like structure on the vertical direction (Figure 2(a3)). As shown by the plasma beta (Figure 2(b3)), the magnetic field is enhanced to $\beta \sim 0.1$ by compression of the gas in a few rotation periods. However, re-amplification to $\beta \sim 0.01$ occurs by no compression effects. We will recall this to show in Figure 5.

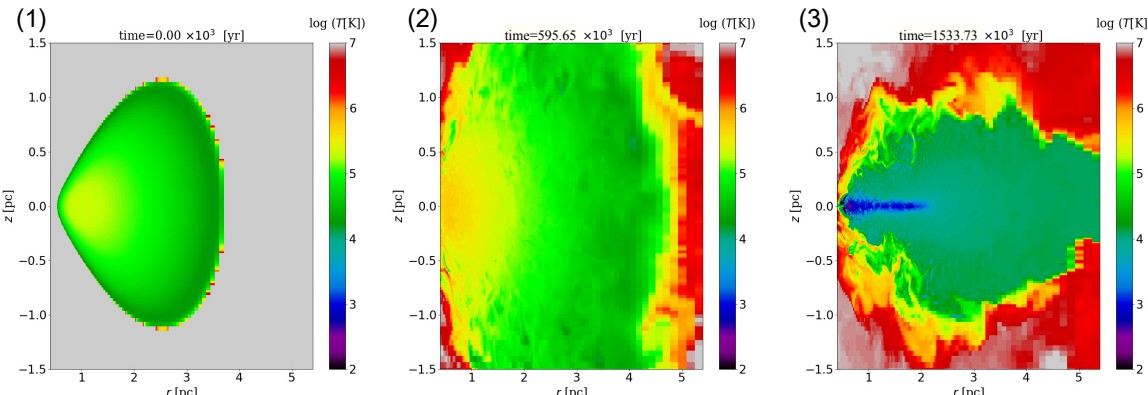

**Figure 1.** Time evolution of the model of $N_\varphi = 512$ in the slice of rz-plane. Color contour denotes temperature. Snapshots are (**1**) $t = 0.00$ [Myr], (**2**) $t = 0.60$ [Myr], and (**3**) $t = 1.53$ [Myr].

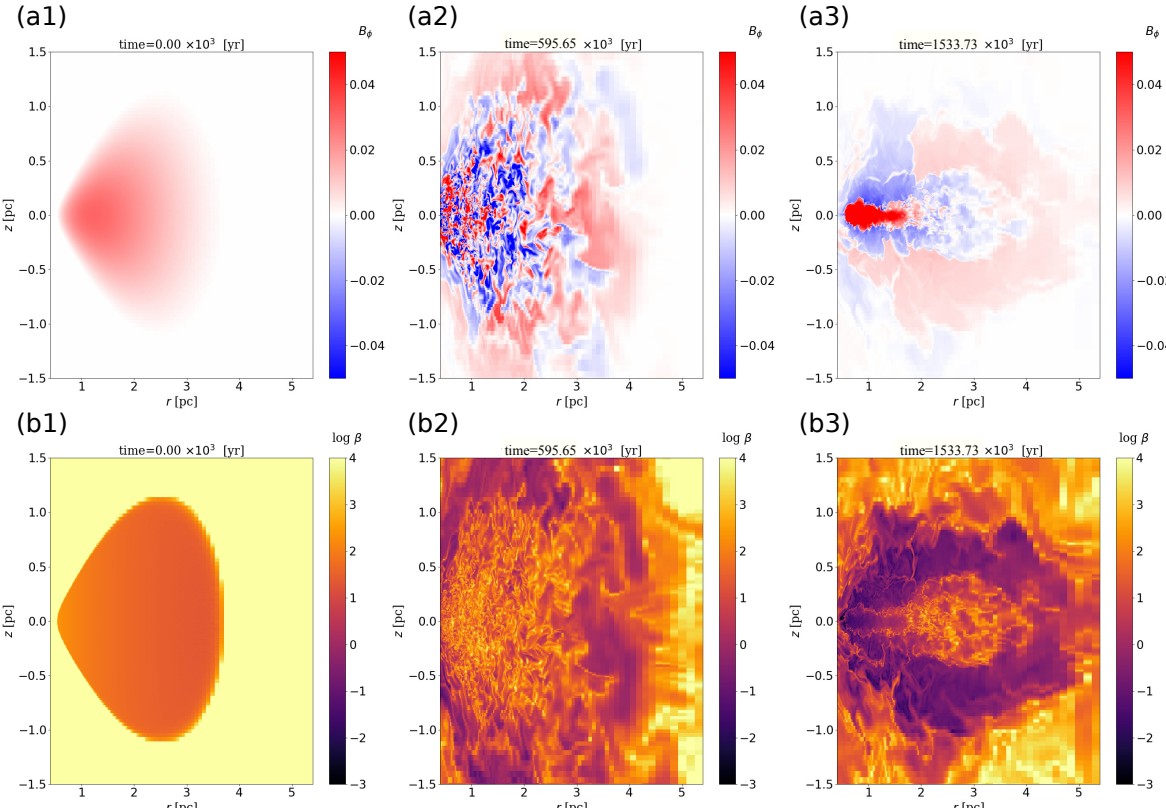

**Figure 2.** The same of time snapshots as Figure 1 but for the distribution of $B_\varphi$ (**top panels**) and plasma $\beta$ (**bottom panels**).

In the cold, thin disk seen in Figures 1(3) and 2(a3), the turbulent structures are not apparent. We here investigate whether this is due to lack of the numerical resolution for the azimuthal direction

to resolve MRI. Hawley et al. [25] suggested that Q-value, which is the ratio of grid size to the characteristics MRI wavelength,

$$Q_\varphi = \frac{\lambda_{\mathrm{MRI}}}{r\Delta\varphi} = \frac{2\pi}{\Delta\varphi}\frac{|B_\varphi|}{v_\varphi\sqrt{4\pi\rho}},\tag{12}$$

should be large enough to resolve the MRI, e.g., $Q_\varphi \geq 20$. Figure 3 shows the $Q_\varphi$ distribution in the two models with different spatial resolutions. Low resolution model ($N_\varphi = 128$) in the top of Figure 3 holds the region of $Q_\varphi < 20$ regardless of whether the radiative cooling and heating are effective or not. On the other hand, $Q_\varphi > 20$ in most regions in the high resolution model ($N_\varphi = 512$). However, one should note that the MRI may be still not well resolved in the cold disk ($1 < r\,[\mathrm{pc}] < 4$), where $Q_\varphi \sim 4$–15.

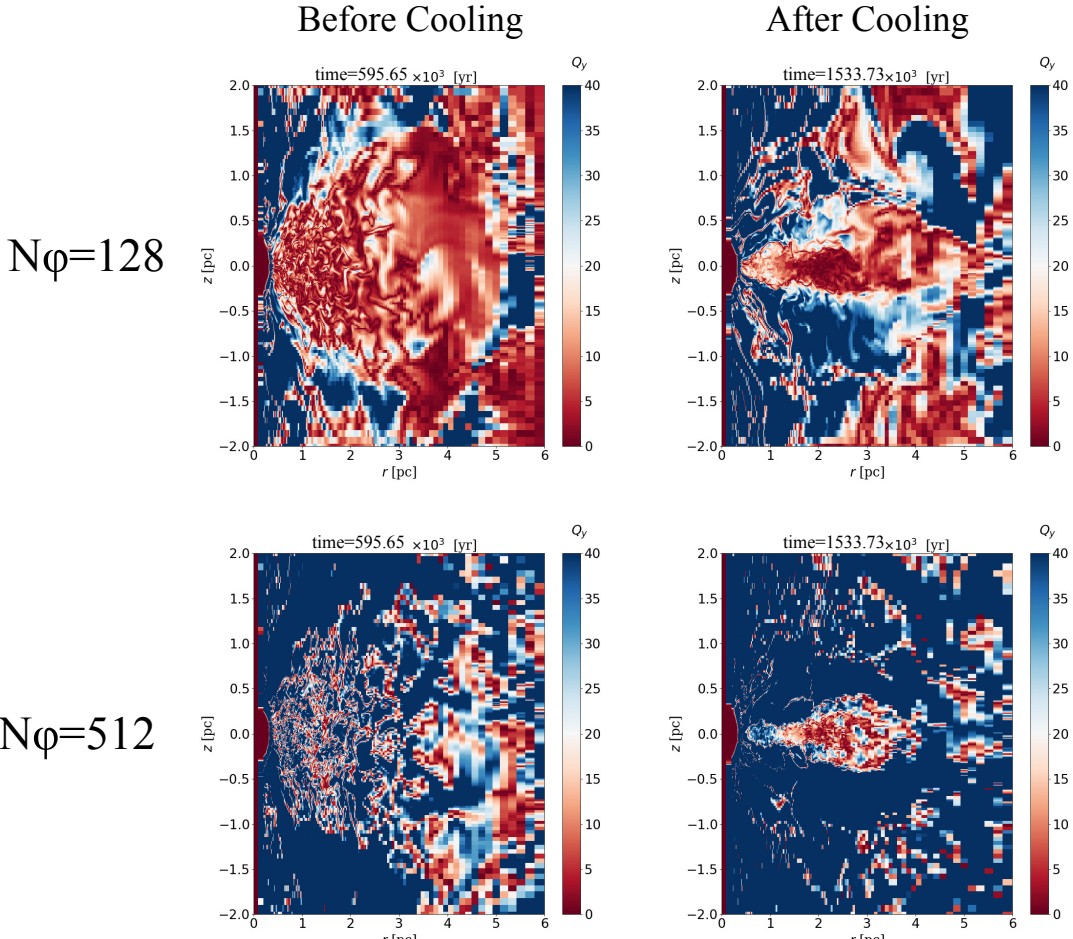

**Figure 3.** Distribution of Q-value defined as the resolution of typical wave length growing MRI. The **left panel** denotes the low azimuth-resolution $N_\varphi = 128$, and the **right panel** denotes the high azimuth-resolution $N_\varphi = 512$. The criteria for resolution is selected by $Q_\varphi = 20$, shown in [25].

Time evolution of azimuthally averaged $B_\varphi$ at $r = 1\,[\mathrm{pc}]$ is shown in Figure 4. Red and blue colors represent that the direction of $B_\varphi$ is opposite. Before the heating and cooling tern on ($t < 0.60\,[\mathrm{Myr}]$), the direction of $B_\varphi$ is varied from positive to negative inside the torus. This implies that the inner magnetic field escapes buoyantly from the torus in the vertical direction due to Parker instability. Although the characteristic timescale growing linear MRI is the rotational period, the timescale of the direction reversal appears about 10 rotational periods, $T_{\mathrm{cycle}} \sim 2.98 \times 10^{-2}\,[\mathrm{Myr}]$ at $r = 1\,[\mathrm{pc}]$. This implies that the quasi-periodic reversal is dominated by the nonlinear growth of MRI, as seen

in the simulations of the galactic or accretion disks (e.g., [5–7,26,27]). When the heating and cooling are taken into account ($t > 0.60$ [Myr]), the difference between the two models is evident. In the high resolution model, the direction of $B_\varphi$ at the mid-plane is reversed, as shown by the change of color from blue to red, at $t \sim 2.0$ [Myr]. This shows that the magnetic field escapes from the disk plane vertically. However, the low resolution model does not begin to occur the reversal. The difference of the magnetic field structure is responsible for the azimuthal resolution.

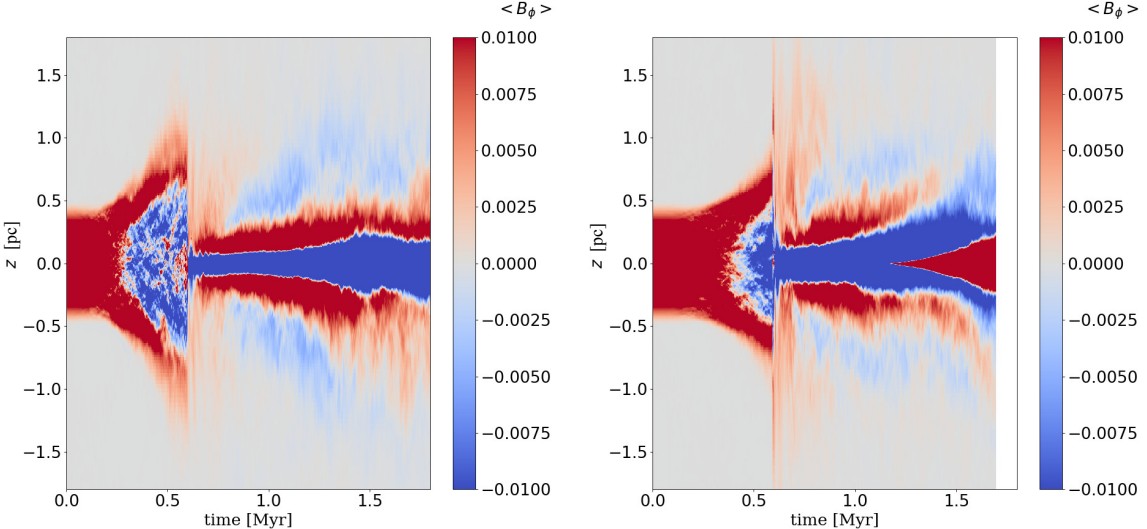

**Figure 4.** Butterfly diagram: time evolution of $B_\varphi$ averaged by azimuthal direction at $r = 1$ [pc]. Color contour of red and blue denotes the different sign of $B_\varphi$. The **left and right panels** denote the low resolution $N_\varphi = 128$ and high resolution $N_\varphi = 512$, respectively. Time evolutions from the adiabatic phase to the cooling/heating phase are switched with $t \sim 0.60$ [Myr].

We reveal the relation between the magnetic field structure and the spatially spread due to the gas motion. We show the 2D histograms of the total magnetic field strength and number density in Figure 5. The color contour denotes the mass occupied in the cell, $\int M(\rho, B_{\text{total}})\, d\rho dB_{\text{total}}$. Left panels are the snapshots of $t = 0.60$ [Myr] before heating and cooling are included. Most of the gas is in the regime where the total magnetic field strength is proportional to the number density on the slope, $B_{\text{total}} \propto n$. Assuming the conservation of mass and magnetic flux in the torus, this relation means that magnetized torus is spread-out in the $rz$-plane, as shown in Figure 1(1) and Figure 2(1b). Compression of the cold gases under the cooling effects is shown in middle panels ($t = 1.10$ [Myr]). The maximum number density increases up to $> 10^4$ [cm$^{-3}$], and the field strength is amplified. In the right panels ($t = 1.53$ [Myr]), the low resolution model ($N_\varphi = 128$) indicates that the field strength does not depend on the number density, $B_{\text{total}} \propto n^0$. This means that the magnetic field structure leaves unchanged. This is agreement with the left panel of Figure 4. On the other hand, the high resolution model ($N_\varphi = 512$) forms the field amplification with the relation, $B_{\text{total}} \propto n^{1/2}$. The right side of maximum field strength is stronger than the middle by a factor of $\sim 2$. The higher the resolution, the stronger the magnetic field generated in the dense region ($1 < n < 4$).

We measure the inflow rate evaluated with the mass flux passing through the cylindrical surface at each radius (Figure 6). Both models show that the inflow rate at $r = 1$ [pc] increases, starting from 1.00 [Myr] ($N_\varphi$=128) or 1.20 [Myr] ($N_\varphi = 512$). The inflow rate is larger with the higher spatial resolution. After $t = 1.50$ [Myr], the inflow rate at each radius decreases in the low resolution model. On the contrary, the high resolution model only shows that the larger the radius, the later inflow rate changes. These results suggest that the angular momentum transport due to the MRI-driven

turbulence, and the resultant mass accretion toward the center, are not well resolved for the model with $N_\varphi = 128$.

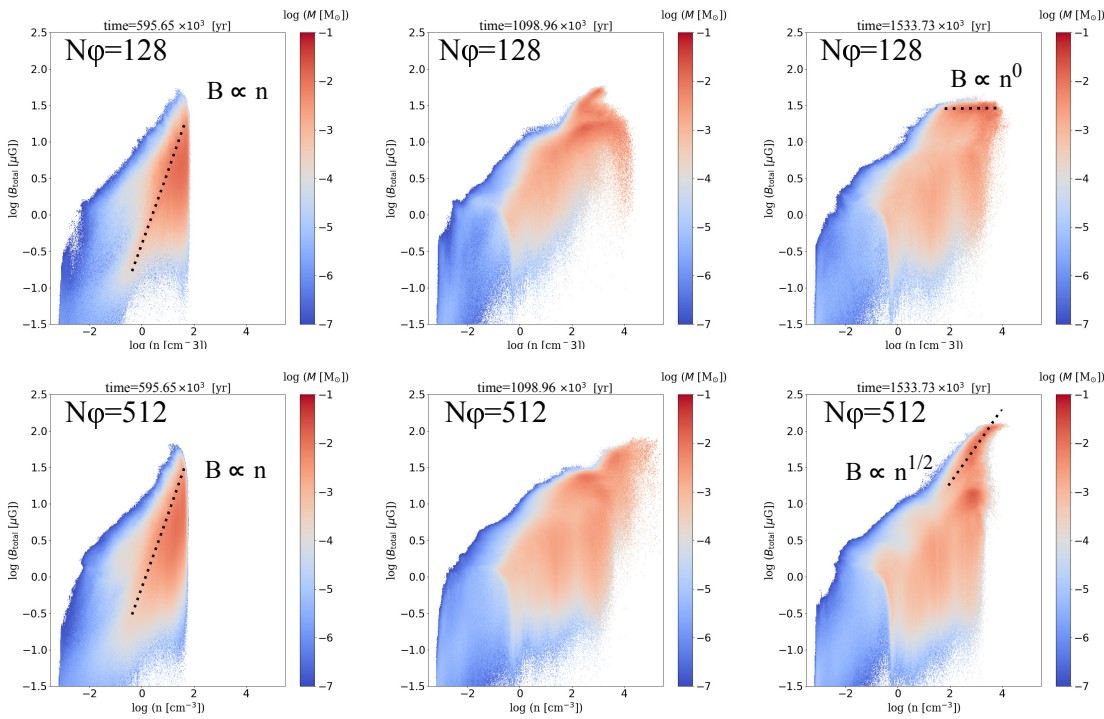

**Figure 5.** 2D histogram of total field strength $B_{\text{total}} = \sqrt{B_r^2 + B_\varphi^2 + B_z^2}$ and number density $n$. Color contour denotes the mass occupying cells of $\Delta(\log n) = \Delta(\log B_{\text{total}}) = 0.01$. **Top panels** show the low resolution model and the **bottom panels** show the high resolution model. **Left**: $t = 0.60$ [Myr] of 20 rotation periods in the pure MHD; **Middle**: $t = 1.10$ [Myr] of 37 rotation periods in the MHD including cooling and heating effects; **Right**: $t = 1.53$ [Myr] of 52 rotation periods.

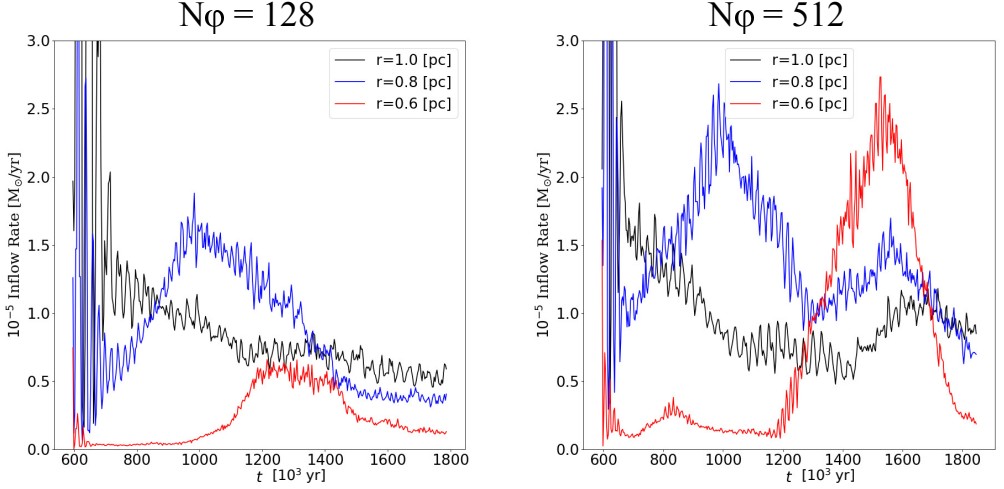

**Figure 6.** Inflow rate after cooling. The **left panel** is the model of $N_\varphi = 128$, and the **right panel** is the model of $N_\varphi = 512$. Red, blue, and black curves denote the radius of $r = 0.6, 0.8$, and $1.0$, respectively.

## 4. Conclusions

We performed three-dimensional MHD simulations including heating and cooling effects in the gas around an AGN. We especially focused on the development of the MRI in a cold gas disk.

The MRI-driven turbulence is developed and it makes the disk geometrically thick. We found that the azimuthal numerical resolution affects driving MRI significantly. For example, quasi-periodic reversal of the mean azimuthal magnetic field does not occur in the low-resolution model, where the Q-value (Equation (12)) is less than about 20 in the whole region. We confirmed that $Q_\varphi$ should be larger than 20, in order to resolve the MRI and buoyantly escape vertically due to Parker instability.

**Author Contributions:** Y.K. performed the simulation and analyzed the data. Both authors contributed to writing and proofreading the article.

**Funding:** This work was supported by JSPS KAKENHI Grant No. 16H03959.

**Acknowledgments:** We thank the anonymous reviewers, Ryoji Matsumoto, and Mami Machida for their helpful comments. This work was supported by JSPS KAKENHI Grant No. 16H03959. Numerical computations were carried out on Cray XC30 and XC50 at the Center for Computational Astrophysics, the National Astronomical Observatory of Japan. Visualization was performed using Python at https://www.python.org/.

**Conflicts of Interest:** The authors declare no conflict of interest.

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
