# Peer review of "Magneto Rotational Instability in Magnetized AGN Tori"

_galaxies, doi:10.3390/galaxies6040139_

Reviewer 1 Report

Nice work! This paper explores the effect of numerical resolution in simulating accretion disks, looking into a number of known features. It is a good idea do such tests and see if the MRI is properly resolved, especially since a number of simulations in the literature have been comparable to the "low resolution" runs, which do not resolve the MRI properly.

For publication, I would like to see: 

* A bit more detail / a reference for the initial condition used, and the type of disc that gets set up. E.g., you are not in the MAD regime, right?

* Do you think N=512 is converged in some sense (e.g. some global properties may be formally converged) or will B-field amplification increase with more resolution? Any way to test this? I understand that it might not be feasible to go to higher resolution right now. But it would be interesting to see if an N=256 simulation has accretion rates (fig 6) are similar to 512.

A caution to the authors -- it is unknown if the Dedner cleaning scheme may articially boost the B-field in such a complicated flow. Using a constrained transport discretization scheme in the future, if possible, is preferred, which preserves div B by construction.

There are also some typos to correct:

* implemented by the HLLD method --> implemented with the HLLD method

* The numbers of grid are --> The numbers of grid points are

* the outflow boundary and absorbing boundary are --> outflow boundary conditions and absorbing boundary conditions are

* heating terms are taken account --> heating terms are taken into account

* The inflow rate is larger in the high with the higher spatial resolution ?word missing? 

Author Response

Thank you for your comment.  We have attavhed the pdf-file with the revised manuscript.  The revised parts are highlighted by the red color.

Reviewer 2 Report

The authors show new 3D simulations of the MRI in AGN tori including heating and cooling. While this is an interesting and important topic, the paper needs to improved in mainly two aspects.

First, the paper show put better in the scientific context. There are a number of recent publications that dealt with accretion disks / tori and the MRI and the authors must discuss differences and the relevance of their own result in respect to these other papers. In particular I mention Gammie et al 2003, Noble et al 2009, Noble et al. 2010. A more recent work is Qian et al. 2017 who essentially investigated how the magnetic diffusivity governs the MRI. Note that the papers just mentioned are general relativistic MHD simulations and the authors should thus discuss the validity of their non-relativistic simulations in the view of existing GR-MHD simulations.

Secondly, the authors must explain in more detail their model approach. What is the exactly initial condition, there is no equation given? Is it stable, did they test its stability with their own code? If not, the instability they detect may not be the MRI. Actually the figures they present for intermediate time scales look like that  the initial torus is not stable, as it is disrupted completely. What is the initial plasma-beta? What is the resistivity eta they apply? This is never mentioned. (note that eta is used for the resistivity (eq. 6) and also for the accretion efficiency (eq.1).

They authors provide physical units as years, central mass Msun, pc, etc. How do they normalize their equations in respect to the torus density? How do they normalize their accretion rate?

I'm somewhat surprised about  the long time scale of Myrs. I would expect the gas is rotating much faster around a black hole.

In the conclusions the authors state that the MRI makes the disk geometrically thick. I would expect instead that the disk becomes flat (see the above mentioned papers), as angular momentum is redistributed, and material becomes accreted. So why is the disk expanding so strongly?

Author Response

Thank you for your comments.  We have attached the pdf-file for the comments and revised manuscript.  The revised parts are highlighted by the red color.
